# Relationship between Climate-Shaped Urbanization and Forest Ecological Function: A Case Study of the Yellow River Basin, China

Xiaobing Gu [1,2], Guangyu Wang [3], Shunli Zhang [1,2], Linyan Feng [4], Ram P. Sharma [5], Huoyan Zhou [6], Liyong Fu [4], Qingjun Wu [1,2], Yaquan Dou [1] and Xiaodi Zhao [1,3],*

1   Research Institute of Forestry Policy and Information, Chinese Academy of Forestry, Beijing 100091, China; guxiaobing@cau.edu.cn (X.G.); bs20203110778@cau.edu.cn (S.Z.); wqj2020@cau.edu.cn (Q.W.); douyq@caf.ac.cn (Y.D.)
2   College of Economics and Management, China Agricultural University, Beijing 100083, China
3   Faculty of Forestry, The University of British Columbia, Vancouver, BC V6T 1Z4, Canada; guangyu.wang@ubc.ca
4   Research Institute of Forest Resource Information Techniques, Chinese Academy of Forestry, Beijing 100091, China; linyan_feng@caf.ac.cn (L.F.); fuly@ifrit.ac.cn (L.F.)
5   Institute of Forestry, Tribhuvan University, Kathmandu 44600, Nepal; ramsharm1@gmail.com
6   School of Ecology and Environment Science, Yunnan University, Kunming 650031, China; zhouhuoyan@itc.ynu.edu.cn
*   Correspondence: zhaoxiaodi@caf.ac.cn; Tel.: +86-136-7105-3260

**Abstract:** Amidst the compounded challenges posed by global climate change and urbanization on forest ecosystems, the integration of urbanization control measures within a climate-focused framework may offer an avenue for breakthroughs. This study delves into the impact of climate, specifically hydrothermal conditions, on the complex interplay between urbanization (Urb) and forest ecological function (Eco) in the Yellow River Basin (YRB) in China. Our findings reveal: (1) The application of a coupled coordination model reveals a stronger alignment between urbanization and forest ecological function in the warm and humid regions in the YRB. (2) Through the cross-sectional threshold regression model, we elucidate the diverse responses of Urb to Eco across varying climate gradients. Among them, annual precipitation shows a double-threshold effect at 532.34 mm and 694.18 mm. As precipitation increases, the impact of Urb shifts from negative to positive on Eco. Moreover, in regions with precipitation below 532.34 mm and above 694.18 mm, the absolute value of response coefficients of Eco to Urb is amplified. Annual average temperature displays a single-threshold effect at 10.11 °C, leading to a transition from negative to positive impact as temperature rises. This study establishes the climate-based threshold system that governs the urbanization–forest ecological function relationship.

**Keywords:** urbanization; forest ecological function; climate; coupled coordination model; threshold regression model

## 1. Introduction

Climate plays a paramount role in shaping the characteristics and geographic distribution of forest ecosystems [1]. Over the past century, the Earth's climate has been undergoing a significant transformation, primarily characterized by global warming [2]. Compounded by the effects of urban development, the tension between preserving forest resources and the growing land requirements for urban expansion is becoming increasingly conspicuous [3]. Extent-wise, with urban expansion, the substantial reduction in forest cover becomes highly evident, thereby resulting in the diminishing ecological functions of regional forest ecosystems [4]. Demographically, the concentration of people in urban areas significantly amplifies the demand for forest resources, such as timber and water,

thus intensifying the ecological pressure on forests [5]. Industrially, non-agricultural sectors further elevate the ongoing demand for forest resources [6]. Spatially, despite alterations in land use due to urban spatial expansion, it is plausible that urban green coverage may not adequately correspond to these changes [7]. Therefore, it is imperative to promptly implement the corresponding environmental and social governance measures to mitigate the adverse impacts that climate change and urbanization may exert on forests.

Compared to other ecosystems, the forest ecosystem exhibits a reduced capacity for adaptation and a slower response to the impacts of climate change [8]. Managing the urbanization process in line with climate conditions may offer a potential solution to the ongoing conflicts between urbanization and forest ecology. However, the influence of urbanization on forest ecology is intricate and multifaceted. Studies in this area can be categorized into two primary groups. The first category of research contends that urbanization has had detrimental effects on forest ecology. The expansion of urban areas, along with high-intensity human activities and irrational land use practices, have resulted in several adverse consequences for forest ecology. These include a decline in ecological function, a reduction in biodiversity, and the destruction of habitats for various plant and animal species [9]. For instance, the study by Deng et al. (2021), which examined natural forests in the Liaohe River Basin in China, revealed that urbanization led to a decrease in the water-holding capacity of the soil layer, the apomictic layer, and the forest canopy [10]. Additionally, Qian et al. (2022) found that key drivers of the spatial and temporal shifts in forest ecological function in Guangzhou, China, between 1979 and 2012 included alterations in forestry policies, urbanization, industrialization, and human disturbances. Of these, excessive logging due to the division of mountains into household plots, forest land encroachment driven by economic overexpansion, and forest degradation all contributed to a decline in the quality of regional forest ecological functions [6].

An alternative body of research believes that urbanization yields a positive impact on forest ecology. The factors contributing to this shift in the impact direction can be summarized as follows: Firstly, the development of urbanization is often accompanied by technological innovations and the adoption of green technologies, reducing cities' reliance on natural resources [11]. This fosters a more intensive development mode, leading to a substantial increase in production efficiency. Ehrhardt-Martinez et al. (2002) found that during the initial stages of urbanization, the deforestation rate continued to rise. However, as urbanization progresses, labor productivity and the environmental consciousness of residents improve, leading to a decline in the rate of destruction and consumption of natural resources, such as forest land [12]. Secondly, the growing global environmental awareness has the potential to instigate more stringent environmental measures. Through the agglomeration effect, urbanization fosters the dissemination of knowledge and skill, as well as the sharing of public resources, thereby enhancing the efficacy of centralized environmental governance [7]. Song et al. (2018) pointed out that environmental education and public engagement could prompt cities to actively invest in their ecological environments [13]. Thirdly, the positive impact of urbanization is interconnected with climate conditions [14]. Climate comfort significantly influences gross regional product growth and population expansion [15], albeit across different stages. Wen et al. (2018) discovered that precipitation plays a substantial positive regulatory role in the process of urbanization affecting vegetation cover [16]. However, Li et al. (2021), based on the Beijing–Tianjin–Hebei region, found that the level of urbanization had no significant impact on the degree of forest fragmentation [17].

The aforementioned studies indicate that there is no consistent consensus regarding the relationship between urbanization and forest ecology, and the influence of climate in this context still remains ambiguous. To illustrate this, this study utilizes the Yellow River Basin (YRB) in China as a case study. It explores the relationship between urbanization (Urb) and forest ecological function (Eco) by examining the coupling coordination level and impact coefficient, respectively, while considering the influence of climate. The specific research ideas are outlined as follows: (1) Employing a coupled coordination model to

investigate the developmental status of the "Urb-Eco" system in the YRB. This model enables an exploration of the degree of coordination between urbanization and forest ecological function. (2) Utilizing the cross-sectional threshold model to assess the nonlinear impact of urbanization on forest ecological function. This analysis aims to identify the changing patterns of urbanization's effect on forest ecological function under varying climate gradients, providing valuable insights into the dynamic nature of this relationship.

At the county level, this study conducts a quantitative analysis of the interplay between forest ecology and urbanization in the YRB, encompassing different regions within the basin. By doing so, the research elucidates the existing conflicts between urbanization development and ecological environmental preservation in the YRB and its adjacent areas. This analysis serves as a foundation for understanding and addressing the challenges arising from the intersection of urbanization and forest ecology in the context of global climate change. Additionally, the findings of this study are expected to inform the formulation of pertinent environmental and socio-economic governance policies, offering practical solutions to the complex dilemma between urbanization and forest ecology.

## 2. Materials and Methods

### 2.1. Study Area

The YRB (Figure 1) spans from 95°53′ E to 119°12′ E in longitude and from 32°9′ N to 41°50′ N in latitude, covering an area of $7.52 \times 10^5$ km$^2$ [18]. This basin encompasses nine provinces and autonomous regions, including Qinghai, Sichuan, Gansu, Ningxia, Inner Mongolia, Shaanxi, Shanxi, Henan, and Shandong. The predominant land use types are grasslands and agricultural land, followed by unused land, forests, and water bodies [19]. Due to the influence of atmospheric circulation and monsoon patterns, there are significant climate variations across different regions within the basin, leading to uneven seasonal distribution [20]. The northwest experiences arid climate conditions, the central region has a semi-arid climate, while the southeast enjoys a semi-humid climate [21]. The annual average temperature ranges from −4 °C to 14 °C, with notable variations and west-to-east gradient from cold to warm [20]. Precipitation is concentrated and unevenly distributed, with an average annual rainfall of 470 mm [22]. Low and uneven rainfall, low humidity, and high evaporation rates are the main climatic characteristics of this region, with frequent occurrences of hail and sandstorms.

### 2.2. Research Methods

#### 2.2.1. Coupled Coordination Model

To analyze the coordinated development status of the urbanization and forest ecological function subsystems, and explore their preliminary relationship with climate factors, we attempted to consider Urb and Eco as two subsystems and introduced a coupling coordination model. In this paper, we referred to the practices of Wang Jintao (2019) to calculate the comprehensive score, coupling degree, and coupling coordination degree [23], which involves the following steps:

First, calculating the comprehensive score, $T_i = \alpha Eco_i + \beta Urb_i$, where $\alpha$ and $\beta$ represent the weights of the comprehensive evaluation indexes for the forest ecological function subsystem and urbanization, respectively. Both $\alpha$ and $\beta$ were set to 0.5. Second, calculating the coupling degree, $C_i = \sqrt{Eco_i \times Urb_i / (\frac{Eco_i + Urb_i}{2})^2}$, where $C_i$ represents the overall coupling degree of the two subsystems in county i. The coupling degree values range between 0 and 1. $C$ equals 1 when the coupling degree is at its maximum, indicating the highest level of interaction and mutual influence between the systems and moving towards new and more advanced functional states. On the other hand, $C$ equals 0 indicating no interaction or mutual influence between the systems, tending towards disordered development. Third, calculating the coupling coordination degree, $D_i = \sqrt{C_i \times T_i}$. The larger the value of D, the higher the coupling coordination degree, indicating not only an overall improvement in the composite system, but also a more coordinated coupling relationship. We classified

the sample counties based on the Urb-Eco coupling coordination degree according to the classification criteria [24] in Table 1.

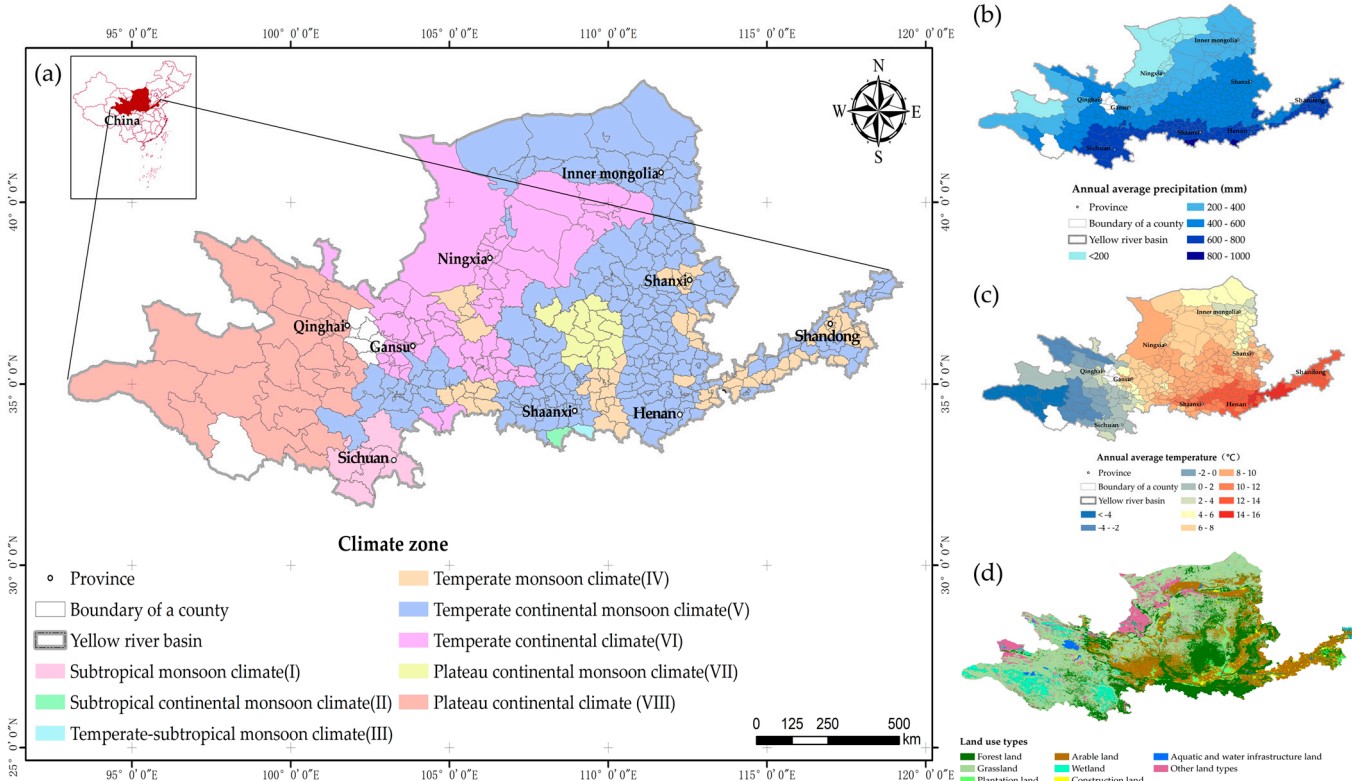

**Figure 1.** The overview of the Yellow River Basin (YRB): (**a**) climate zone; (**b**) annual average precipitation; (**c**) annual average temperature; (**d**) land use types. Note: climate type data for each county were collected from the local government's official websites; see Table S1 for more specific information. Annual average precipitation and annual average temperature data were obtained from the "WorldClim 2.1 Climate Data (1970–2000)" published by the WorldClim database (https://www.worldclim.org, accessed on 29 September 2023) and land use type data were provided by the National Forestry and Grassland Administration Planning Institute.

**Table 1.** Grade criteria of coupling degree and coupling coordination degree.

| C Grade Criteria | [0.3,0.5) | | [0.5,0.8) | | [0.8,1.0] |
| --- | --- | --- | --- | --- | --- |
| | Antagonistic Stage | | Adjustment Stage | | High Coupling Stage |
| D Grade Criteria | [0,0.1) Extremely imbalanced | [0.1,0.2) Seriously imbalanced | [0.2,0.3) Moderately imbalanced | [0.3,0.4) Slightly imbalanced | [0.4,0.5) Nearly imbalanced |
| | [0.5,0.6) Barely coordinated | [0.6,0.7) Primary coordinated | [0.7,0.8) Intermediate coordinated | [0.8,0.9) Good coordinated | [0.9,1.0] High-quality coordinated |

C refers to the coupling degree and D refers to the coupling coordination standard degree.

### 2.2.2. Cross-Sectional Threshold Model

To investigate the moderating effect of climate on the impact of urbanization on forest ecological function, this study employed Hansen's cross-sectional threshold model. Precipitation and temperature were used as threshold variables separately. This study explored the nonlinear influence of urbanization on the forest ecological function and further determined the relationship between these factors and the role played by climate variables.

Artificially defining the gradients and introducing interaction or quadratic terms for explanatory variables are common methods for performing grouping heterogeneity analyses. However, most of these methods had shortcomings, such as inconsistent grouping criteria, subjectivity, unclear interpretability, and an inability to test significance [25,26]. Hansen's threshold model was developed to address these issues. This model selects the number and values of thresholds "endogenously", thus substantially reducing the subjectivity associated with artificial grouping [27]. Moreover, the existence of the threshold effect and the authenticity of threshold values can be verified [27]. Without loss of generality, the construction of the cross-sectional multi-threshold model is as follows:

$$
\begin{aligned}
\text{Eco}_i = \beta_0 + \beta_1 \cdot Urb_i \cdot I_i(Thres_i \le \lambda_1) + \beta_2 \cdot Urb_i \cdot I_i(\lambda_2 < Thres_i \le \lambda_2) + \cdots \\
+ \beta_k \cdot Urb_i \cdot I_i(\lambda_k < Thres_i \le \lambda_k) + \gamma_i Control_i + \varepsilon_i (k < 436)
\end{aligned} \tag{1}
$$

where: $i$ represents county-level variables; Eco represents forest ecological function scores; *Urb* represents the level of urbanization; *Control* represents a series of control variables; *Thres* represents threshold variables, specifically set as the annual average precipitation (Pre) and the annual average temperature (Temp); *I* is an indicator function, essentially a segmented function related to the threshold variable *Thres*. It takes the value 1 when the threshold variable is within that interval; otherwise, it takes 0. $\lambda$ is the estimated threshold value, $\beta$ is the coefficient estimated using the nonlinear least square method, $k$ is the number of thresholds, $\gamma$ is the coefficient of each control variable, and $\varepsilon$ is the random disturbance term in the model.

The testing of the threshold model primarily consists of two parts [27–29]. First is the F-test for the existence of threshold effects, that is, the significance test of threshold effects, with the null hypothesis $H_0 : \beta_1 = \beta_2 = \cdots = \beta_k$. The F-statistic is $F = (S_0 - S_k(\hat{\lambda}))/\hat{\sigma}^2$, where $S_0$ is the sum of squared residuals when there is no threshold value. Since the threshold value $\lambda$ cannot be identified under the null hypothesis $H_0$, the distribution of the F-statistic is non-standard [28]. Therefore, Hansen (1999) proposed the "Bootstrap" resampling method to obtain the asymptotic distribution of the F-statistic and construct the corresponding *p*-value for significance testing [30]. This test assesses whether there would be significant differences in the parameters among the $k + 1$ sample groups divided by $k$ threshold values. Second, the LR-test for the validity of threshold values, that is, using maximum likelihood estimation to test whether the estimated threshold value $\hat{\lambda}$ significantly matches the true value $\lambda$, with the null hypothesis $H_0 : \hat{\lambda} = \lambda$. LR-statistic is $LR_k(\lambda) = (S_k(\lambda) - S_k(\hat{\lambda}))/\hat{\sigma}^2$. When $LR_k(\lambda) \le -2\ln(1 - \sqrt{1 - \alpha})$, the null hypothesis cannot be rejected in a condition when $\alpha$ is the significance level for the test of validity.

*2.3. Variables*

2.3.1. Dependent Variable

Forest Ecological Function Index (Eco): Forest ecological function refers to the ecological environment and benefits provided by a forest ecosystem through its inherent ecological characteristics and processes [31–34]. It is conducive to human survival and progress and encompasses a wide range of aspects, including water conservation, soil and water preservation, climate regulation, water purification, and the protection of biodiversity [35]. We constructed Eco based on the existing practices [31–36], following the guidelines outlined in "Technical Regulations for National Forest Resource Continuous Inventory" (GB/T 38590-2020) [37]. The Eco subsystem includes eight components: forest biomass, naturalness, community structure, tree species composition, total vegetation cover, canopy closure, average tree height, and dead leaves thickness. Using the predefined weights, a comprehensive score for this subsystem was calculated. The specific steps are: (1) Each factor is assigned with different scores for the three types (I, II, and III) based on its relevance. These scores for each factor are denoted as $X_i$. (2) Weights ($w_i$) were determined based on their relative importance. (3) The forest ecological function score ($s_F$) for each site was calculated using the formula $s_F = 1/\sum_{i=1}^{8} X_i w_i$. (4) By weighting the site data according to the site area, the site-level data was transformed into the county-level

data, resulting in the comprehensive Eco scores for each sample county in the YRB. For assessment factors of forest ecological function and their classification, please see Table S2 for details.

### 2.3.2. Threshold-Dependent Variable

Urbanization (Urb): Urbanization takes on the multidimensional form of socio-economic change, and a single dimension of the population urbanization is insufficient to reflect the complex characteristics of land use types, migration, industrial structure, residents' lifestyles, and ecological environments [23,38]. Additionally, different dimensions of urbanization representation have varying impacts on ecological efficiency [38]. Therefore, referring to the previous studies [38–42], we evaluated the comprehensive level of urbanization (Urb) in the YRB using indicators from four dimensions: land urbanization (URL), population urbanization (URP), industrial urbanization (URI), and spatial urbanization (URS). We determined the weights of each evaluation indicator using the entropy method. This method calculates the information entropy of the indicator items to determine their weights, making the weight determination more objective [43]. Specifically, we used the proportion of built-up area as a proxy variable for URL, the proportion of the urban population to the total resident population for URP, the proportion of value added of non-agricultural industries for URI, and the green coverage rate of built-up areas for URS.

### 2.3.3. Threshold Variables

The annual average precipitation (Pre) and the annual average temperature (Temp): Hydrothermal conditions exert a significant influence on both the urban and forest ecosystems. Particularly in the region located north of China's Qinling Mountains and Huai River, precipitation has consistently served as a crucial limiting factor for vegetation growth [44–46]. Indeed, climate intricately governs the distribution of forest tree species [47], influences forest productivity [48], shapes forest structure [49], modulates plant functional traits [50], and ultimately impacts the forest ecological functions and services [51,52]. In the previous studies, key explanatory variables related to human socio-economic activities, such as gross domestic product per capita (GDP) [17], degree of openness to foreign trade [53], and urbanization level [38] have been used as threshold variables. However, one of the main objectives of our study was to explore the changes in the relationship between urbanization and forest ecological function under different climatic conditions. Therefore, natural climatic factors were distinguished from the socio-economic activity factors, and hydrothermal conditions were used here as thresholds in our analyses.

### 2.3.4. Control Variables

Within the cross-sectional threshold model, we incorporated controls for pertinent variables that have impacted the forest ecological functions. These variables encompass natural factors, such as altitude, slope, and contiguity level, alongside socio-economic factors including the regional economic scale (GDP), economic density, population density, disposable income of urban residents, and disposable income of rural residents. Moreover, in the domain of human activity factors affecting forest ecological function, it would be crucial not to disregard the policy-related factors. Notably, following the implementation of China's Grain for Green Program, the rate of vegetation recovery has surpassed that of the pre-implementation period by over sixfold [20]. Consequently, our study, in alignment with the actual circumstances in the YRB, considered introducing the three policy-related virtual variables: reforestation intensity, grassland restoration intensity, and wetland restoration intensity. These variables are considered very instrumental in representing the policy impact of significant initiatives in different counties [54,55].

*2.4. Data Sources and Processing*

2.4.1. Data Sources

The data for all variables were obtained from the year 2018, which covers 448 counties in the YRB. The data sources with specific indicators are shown in Table 2.

**Table 2.** Overview of data sources.

| Criteria | Sub-Criteria | Original Data Source |
|---|---|---|
| Natural indicators | Forest biomass; Naturalness; Community structure; Tree species composition; Total vegetation cover; Canopy closure; Average tree height; Litter layer thickness; Altitude; Slope; Contiguous area level; Land use types | The 9th National Forest Resources Inventory in China results, the original data were provided by the Planning Institute of the National Forestry and Grassland Administration. |
| | Climatic zones | Official website of each county's government. |
| | Annual average temperature; Annual average precipitation data | The "WorldClim 2.1 Climate Data (1970–2000)" published by the WorldClim database (https://www.worldclim.org, accessed on 29 September 2023). |
| Socio-economic indicators | Total area by county; Built-up area; Urban population; Year-end resident population; Value added of non-agricultural industry; Regional gross domestic product; Green coverage; Economic density; Disposable income of urban residents; Disposable income of rural residents | The CNKI Big Data Platform (https://data.cnki.net, accessed on 29 September 2023) and national economic and social development statistical bulletins of various counties. |
| | Reforestation intensity; Grassland restoration intensity; Wetland restoration intensity | China Forestry and Grassland Statistical Yearbook 2018 (http://www.forestry.gov.cn, accessed on 29 September 2023) and China Forestry Statistical Yearbook 2002–2017 (https://data.cnki.net, accessed on 29 September 2023). |

2.4.2. Data Processing

For the natural indicators, the county-level data were obtained by applying the area-weighted coefficients based on the sample areas. In cases where certain counties had missing variable values, they were replaced with the averages of the same variable from other counties within the same city. For counties where replacement was not possible, the samples were excluded. After data processing, a total of 436 valid samples were retained, which accounts for 97.32% of the total sample size. Descriptive statistics for the relevant variables are presented in Table 3. Data processing and model analysis were conducted using Microsoft Excel 2010 and StataMP 17, while the visualization of map data was carried out using ArcGIS 10.2.

**Table 3.** Descriptive statistics of variables.

| Variable | Definition | N | Mean | Std. Dev. | Min | Max |
|---|---|---|---|---|---|---|
| Eco | Forest ecological function index. | 436 | 0.438 | 0.065 | 0.342 | 0.691 |
| Urb | Comprehensive urbanization score. | 436 | 0.108 | 0.158 | 0.010 | 1.826 |
| Pre | The annual average precipitation (mm). | 436 | 514.639 | 144.314 | 115.708 | 813.681 |
| Temp | The annual average temperature (°C). | 436 | 8.947 | 4.216 | −4.557 | 15.140 |
| Altitude | Average altitude by county (m). Here, altitude has been log-transformed. | 436 | 6.637 | 1.283 | 0.898 | 8.434 |
| Slope | Average slope by county (°). | 436 | 10.886 | 8.099 | 0.000 | 37.022 |
| GDP | Regional gross domestic product ($10^8$ CNY). Here, GDP has been log-transformed. | 436 | 4.790 | 1.149 | 1.176 | 7.490 |
| Economic density | The ratio of regional gross domestic product to the total area ($10^8$ CNY/km$^2$). | 436 | 0.652 | 2.718 | 0.000 | 41.303 |

**Table 3.** *Cont.*

| Variable | Definition | N | Mean | Std. Dev. | Min | Max |
|---|---|---|---|---|---|---|
| Population density | The ratio of the permanent resident population to the total area ($10^4$ person/km$^2$). | 436 | 0.084 | 0.135 | 0.000 | 0.686 |
| Disposable income of urban residents | ($10^4$ CNY) | 436 | 3.052 | 0.572 | 0.956 | 4.594 |
| Disposable income of rural residents | ($10^4$ CNY) | 436 | 1.268 | 0.431 | 0.377 | 3.352 |
| Contiguous area level | The size of the contiguous area, evaluated according to the forest cover type and takes values from 0 to 7. | 436 | 4.313 | 1.171 | 1.000 | 6.779 |
| Reforestation intensity | A summation dummy variable equals 1 when a county has implemented only one of the projects. including "natural forest resource protection", "grain for green", "construction of key protective forest systems in the Three-North region", "control of wind and sand sources in Beijing and Tianjin", and "construction of fast-growing and high-yield timber forest bases in key areas"; 0 when none of the above projects have been implemented; and the highest value is taken to be 5. | 436 | 3.323 | 0.878 | 1.000 | 4.000 |
| Grassland restoration intensity | A dummy variable equals 1 when a county has implemented the "grazing prohibition and grassland restoration project" and 0 otherwise. | 436 | 0.312 | 0.464 | 0.000 | 1.000 |
| Wetland restoration intensity | A summation dummy variable equals 1 when a county has implemented only one of the projects including the "converting croplands to wetlands" and "wetland protection and restoration"; 0 when none of the above projects have been implemented; and the highest value is taken to be 2. | 436 | 0.789 | 0.750 | 0.000 | 2.000 |

### 2.5. Study Flowchart

Based on the above analyses, we drew a study flowchart (Figure 2).

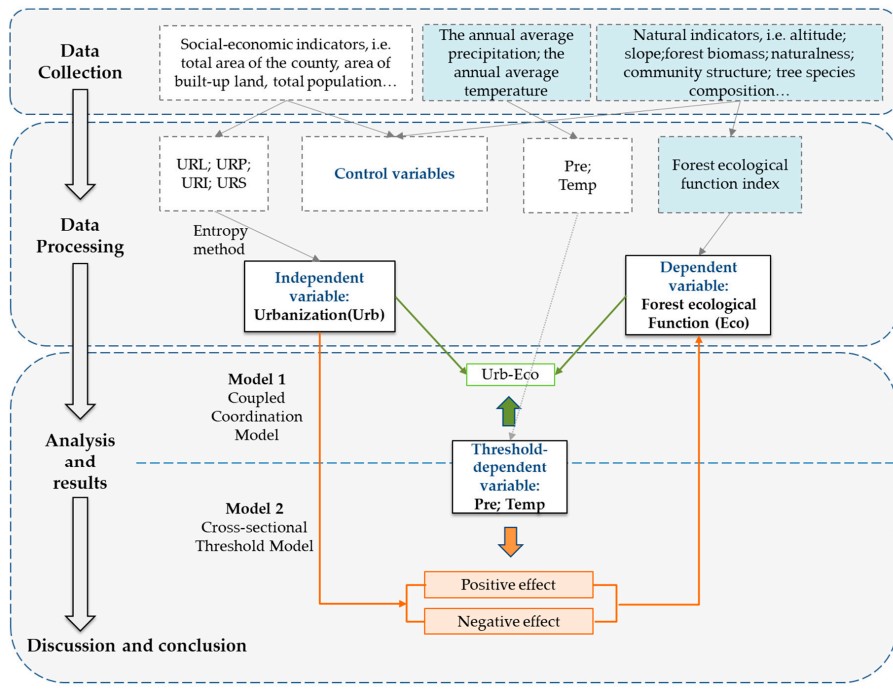

**Figure 2.** A study flowchart. Note: variables in blue boxes are site-level data and variables in white boxes are county-level data, which were weighted and aggregated based on the sample areas.

## 3. Results

*3.1. Coupling and Coordination Results*

3.1.1. Subsystem Scores

The scores of two subsystems, namely the comprehensive urbanization level (Urb) and the forest ecological function index (Eco), are shown in Figure 3.

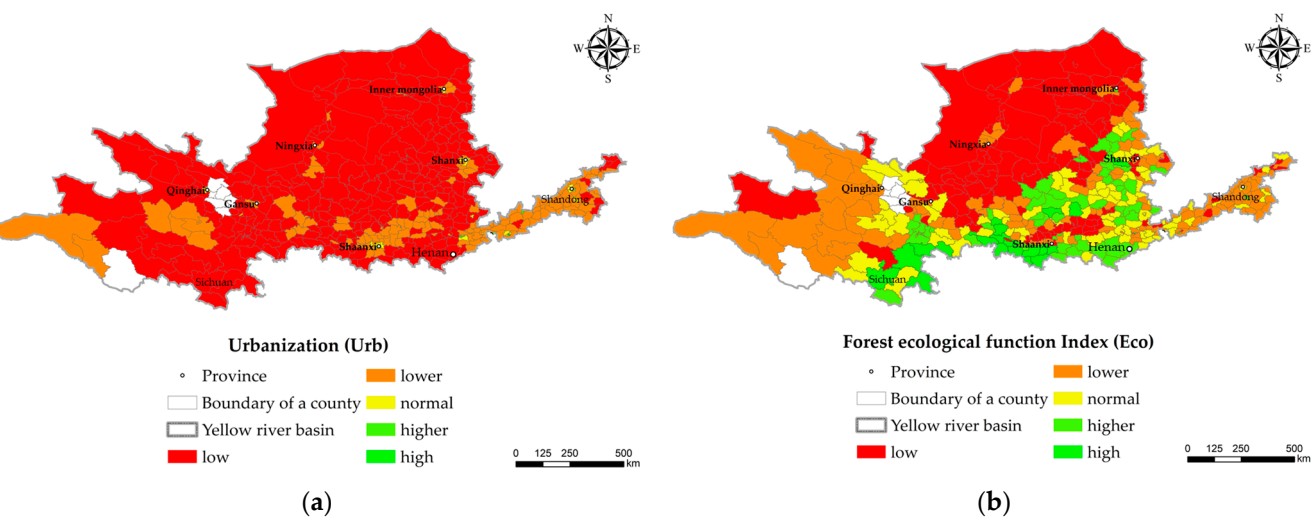

**Figure 3.** Subsystem scores in counties of the YRB calculated by coupled coordination model. (**a**) Subsystem scores for urbanization (Urb); (**b**) subsystem scores for forest ecological function (Eco). Note: each subsystem score is sliced into 5 parts: low, lower, normal, higher, and high, using the natural breakpoint method, respectively.

Figure 3a illustrates the comprehensive scores of the Urb subsystem calculated using the entropy method. In terms of regional distribution, Urb in the YRB shows relatively low overall spatial differentiation, with 75.51% of county-level samples below the average level. Counties with higher Urb scores were only sporadically distributed in the southern part of the YRB. In a county-by-county comparison, Beilin District in Shaanxi province had the highest Urb score, while Alashan Zuoqi in Inner Mongolia had the lowest Urb score.

Figure 3b shows that the forest ecological function scores in the YRB exhibit significant spatial variations. Taking the 400 mm annual precipitation line as a dividing criterion, a basic pattern of "higher in the southeast, lower in the northwest" was observed. Moving from the southern part of the YRB towards the surrounding areas, especially to the north, the forest ecological function gradually weakened. Counties with a "high" score of Eco are located in the southern part of the YRB, where annual precipitation exceeds 600 mm, including southern Shaanxi and Sichuan provinces, with Zhouzhi County in Shaanxi province having the highest Eco score. Counties with a "low" level of forest ecological function had a broader distribution, covering almost the entire YRB region in Inner Mongolia, with the poorest Eco found in Honggu District, Gansu Province. Counties with Eco scores between "high" and "low" show significant differences in the spatial distribution. The counties categorized as "normal" are scattered across various provinces and cities, while those classified as "lower" are mainly found in the upper and lower reaches of the YRB. Counties with a "higher" level are predominantly located in the central-right part of the YRB, primarily in Shaanxi and Shanxi provinces.

Arranging the average scores of each province and municipality, the Urb scores in the YRB can be ranked as follows: Henan > Shaanxi > Shandong > Overall (0.1093) > Qinghai > Gansu > Shanxi > Ningxia > Inner Mongolia > Sichuan. As for the Eco scores, the ranking is Sichuan > Shaanxi > Henan > Shanxi > Gansu > Overall (0.4383) > Qinghai > Shandong > Ningxia > Inner Mongolia. More than half of the provinces scored higher than the YRB's average Eco score. Sichuan province exhibited a significant spatial discrepancy between Urb and Eco: while its Eco score was high, its Urb score was relatively low.

### 3.1.2. Coupling Results

To further explore the relationship between urbanization and forest ecological function, the coupling degree and coupling coordination results of the two were calculated and presented (Figure 4).

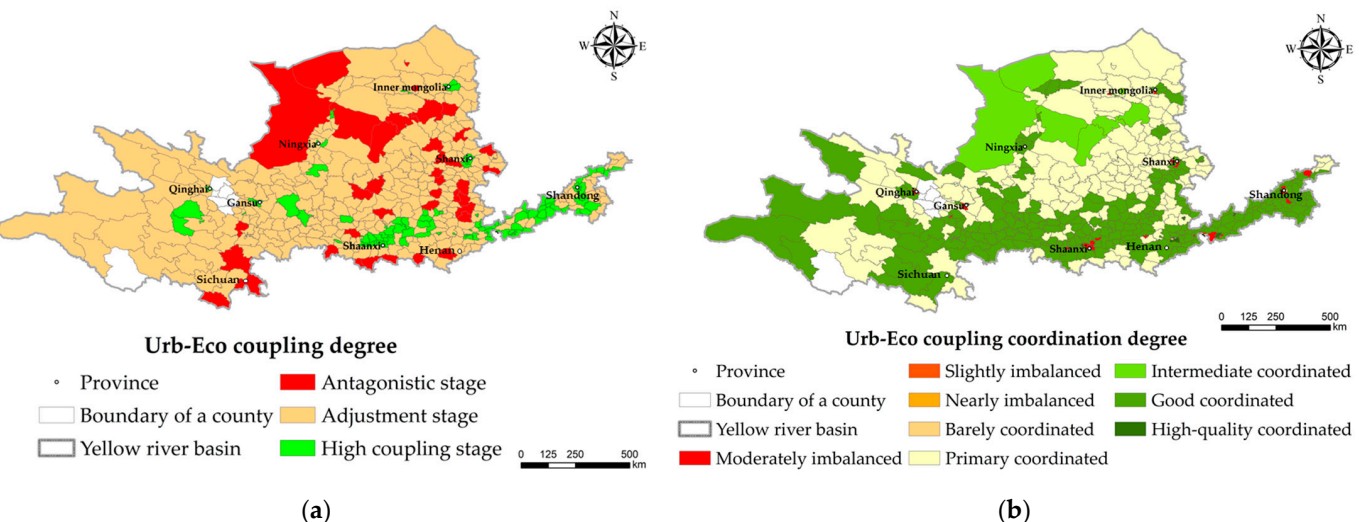

**Figure 4.** Coupling degree and coupling coordination degree of Urb-Eco system in the YRB calculated by coupled coordination model. (**a**) Urb-Eco coupling degree; (**b**) Urb-Eco coupling coordination degree.

Figure 4a shows that most counties are still in the "adjustment stage" of the Urb-Eco continuum. In counties located in the northern part of the YRB, which receives the least annual precipitation, the overall coupling degree was at a relatively low "antagonistic stage". This could be characterized by competition and resource competition between the two subsystems, with alternating dynamics. Counties in the "high coupling stage" are primarily situated in the regions with an annual average precipitation exceeding 600 mm and an annual average temperature above 10 °C. In these areas, there was a high level of interaction between urbanization and the forest ecological function. However, this proportion of counties was relatively low. While coupling degree is important for assessing the strength of coupling interactions and predicting the development order between the Urb and Eco subsystems, it might sometimes fail to reflect the overall functionality or development level of the systems. Therefore, it was necessary to construct a coupling coordination degree model. As evident from Figure 4b, counties with high-quality Urb-Eco coupling coordination are relatively scarce and dispersed, while the majority of counties are still in the "primary coordinated" stage, indicating that there is significant room for improvement in the overall coupling coordination of the system.

Scatter plots and linear regression lines illustrating the relationships between the coupling degree and coupling coordination of the two systems with changes in annual precipitation and annual average temperature are shown in Figure 5. The coordinated development level of urbanization and forest ecological function was positively correlated with both categories of climate factors, indicating that a warm and humid climate favors the coordinated development of urbanization and forest ecological function.

### 3.2. Threshold Model Results

#### 3.2.1. Test of Threshold Effect Existence

Before identifying the threshold effects of the annual temperature and the annual precipitation in the impacts of urbanization on forest ecological function in the YRB, a threshold effect existence test was conducted and the results are presented in Table 4.

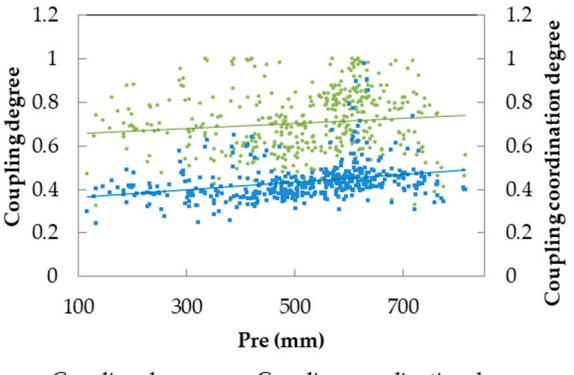
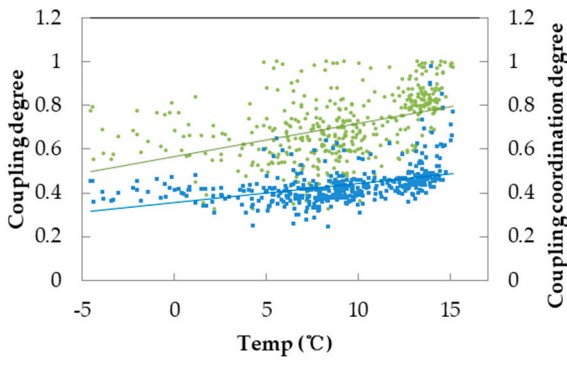

**Figure 5.** Scatter plot and fitting trend line of Urb-Eco coupling degree and coupling coordination degree with the annual average precipitation and the annual average temperature.

**Table 4.** Test results of threshold effect existence.

| Threshold Variable | Threshold-Dependent Variable | Threshold Model Selection | F-Value | *p*-Value | Critical Value | | |
|---|---|---|---|---|---|---|---|
| | | | | | 1% | 5% | 10% |
| Pre | Urb | Single | 26.613 *** | 0.000 | 7.443 | 4.400 | 3.140 |
| | | Double | 6.206 ** | 0.013 | 6.685 | 3.294 | 2.339 |
| | | Triple | 5.671 ** | 0.047 | 9.914 | 5.591 | 3.745 |
| Temp | Urb | Single | 9.730 *** | 0.002 | 7.515 | 4.193 | 2.892 |
| | | Double | 5.578 ** | 0.023 | 8.654 | 4.339 | 3.017 |
| | | Triple | 2.599 | 0.127 | 6.710 | 3.882 | 3.100 |

***, **, and * represent significance at the 1%, 5%, and 10% levels, respectively with 5% being the commonly used significance level. *p*-values and critical values were obtained by simulating 5000 iterations using the Bootstrap method [56].

When taking Pre as the threshold variable and Urb as the threshold-dependent variable, single threshold effects, double threshold effects, and triple threshold effects of Pre were all significant at the 5% confidence level. However, further assessment was needed in conjunction with a test of the authenticity of the threshold values under these three types of threshold models [27,30]. On the other hand, when taking Temp as the threshold variable and Urb as the threshold-dependent variable, the single threshold and double threshold models were significant at the 1% and 5% significance levels, respectively, while the triple threshold model was not significant. Therefore, the triple threshold effect of Temp was preliminarily ruled out.

3.2.2. Test of Threshold Value Authenticity

After detecting the existence of threshold effects, further estimation and validation of the threshold values were conducted. The estimated threshold values and their corresponding confidence intervals under the different threshold models are presented in Table 5. When "Pre" was used as the threshold variable, the estimated value for a single threshold was 532.342 mm, with a 95% confidence interval of [413.342, 559.629]. The estimated values for the second threshold (694.178 mm) and the third threshold (285.953 mm) were both outside of this interval, indicating significant differences between the second and third threshold values compared to the first single threshold value [29]. However, by examining the results of the LR test in Figure 6, the LR values were consistently below the critical threshold (Figure 6d), suggesting that the confidence interval for the third threshold is excessively wide and cannot effectively contribute to the threshold convergence. Therefore, it was eventually decided to adopt the double threshold model, that is, to take 532.342 mm and 694.178 mm as the threshold values of Pre.

**Table 5.** Results of authenticity test of threshold values.

| Threshold Variable | Threshold-Dependent Variable | Threshold Model Selection | The Threshold Estimate | 95% Confidence Interval |
|---|---|---|---|---|
| Pre | Urb | Single | 532.342 | [413.342, 559.629] |
| | | Double | 694.178 | [184.199, 811.225] |
| | | | 532.342 | [413.293, 564.808] |
| | | Triple | 285.953 | [184.199, 748.854] |
| Temp | Urb | Single | 10.105 | [0.238, 13.873] |
| | | Double | 9.191 | [−2.012, 15.092] |
| | | | 10.105 | [9.335, 10.357] |
| | | Triple | 8.942 | [−2.012, 14.570] |

The threshold estimate is the value at which the likelihood ratio test statistic LR = 0.

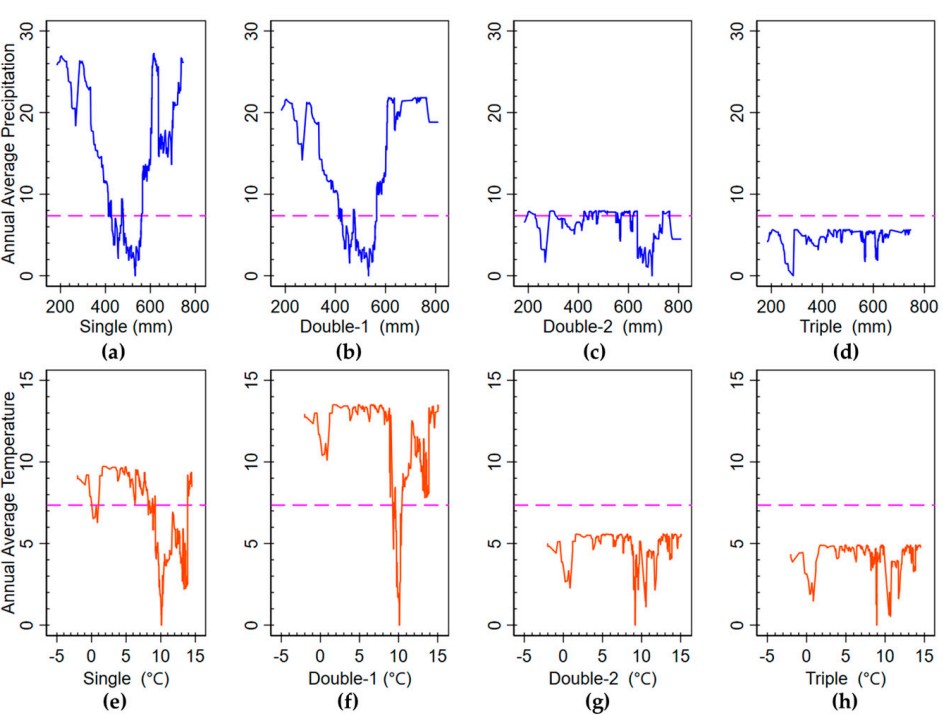

**Figure 6.** Confidence interval construction with each threshold model. (**a**) Likelihood ratio (LR) map corresponding to the single threshold estimate (532.342) for annual average precipitation (Pre). (**b**) LR map corresponding to the first threshold estimate (532.342) under the double threshold model for Pre. (**c**) LR map corresponding to the second threshold estimate (694.178) under the double threshold model for Pre. (**d**) LR map corresponding to the threshold estimate (285.953) under the triple threshold model for Pre. (**e**) LR map corresponding to the single threshold estimate (10.105) for annual average temperature (Temp). (**f**) LR map corresponding to the first threshold estimate (10.105) under the double threshold model for Temp. (**g**) LR map corresponding to the second threshold estimate (9.191) under the double threshold model for Temp. (**h**) LR map corresponding to the threshold estimate (8.942) under the triple threshold model for Temp. Note: in the figure, the dashed horizontal line (LR = 7.352) corresponds to a significance level of 0.05 [27].

Similarly, when "Temp" was considered as the threshold variable, the confidence interval for a single threshold was relatively narrower compared to the confidence intervals for the second and third thresholds, indicating more precise estimation results [28] (Figure 6). Although the double threshold model was significant at the 5% confidence level, the estimated threshold values were considered unreasonable for two main reasons: Firstly, both estimated threshold values of the double threshold fall within the confidence interval of the single threshold. Secondly, the confidence interval for the first threshold value (9.191 °C) of the double threshold model encompasses almost the entire sample

space and exhibits mixed signs at the confidence interval boundary [27]. Since there is no evidence of a triple threshold effect, no further discussion is conducted regarding the third threshold estimation value.

### 3.2.3. Threshold Model Results

The estimation results of the double-threshold model for Pre and the single-threshold model for Temp are presented in Table 6. For the subset with precipitation as the threshold variable (Column 1), using two threshold values, 532.342 mm and 694.178 mm, we divided the samples into three subgroups: $P_1$, $P_2$, and $P_3$. The coefficient of "Urb" on "Eco" changed with an increase in Pre. Initially, it progressively reduced its negative impact, and then it shifted from negative to positive, indicating the ecological promotion effect of urbanization. Specifically, in $P_1$, which mainly includes the county-level samples from Qinghai, Inner Mongolia, and Ningxia, where there are high proportions of the primary industries, which has displaced the forest cover and ecological functions. Additionally, these areas experience the plateau continental climate, temperate continental climate, and temperate continental monsoon climate with distinct seasonal characteristics, including hot and rainy summers and cold, dry winters. Limited precipitation and ecological fragility restrict the growth and recovery of forest vegetation in $P_1$. Consequently, urbanization exerts the most pronounced stress on forest ecological function, with the largest absolute coefficient value of 0.197.

**Table 6.** Regression results of threshold model.

| Variable | (1) | (2) |
|---|---|---|
| Slope | 0.00469 *** (11.00) | 0.00503 *** (11.72) |
| Altitude | −0.00718 ** (−2.03) | −0.00721 ** (−1.97) |
| GDP | −0.00644 ** (−2.02) | −0.00706 ** (−2.19) |
| Economic density | 0.00169 (0.81) | 0.00164 (0.77) |
| Population density | 0.0936 *** (4.10) | 0.0992 *** (4.24) |
| Disposable income of urban residents | −0.0155 *** (−2.95) | −0.0133 ** (−2.47) |
| Disposable income of rural residents | 0.0142 * (1.73) | 0.0121 (1.44) |
| Contiguous area level | 0.00472 * (1.96) | 0.00489 ** (1.98) |
| Reforestation intensity | 0.00509 (1.31) | 0.000283 (0.08) |
| Grassland restoration intensity | −0.0188 *** (−3.14) | −0.0206 *** (−3.38) |
| Wetland restoration intensity | 0.00713 ** (2.06) | 0.00891 ** (2.53) |
| Urb_1 | $P_1$: −0.197 *** (−3.50) | $T_1$: −0.141 ** (−2.38) |
| Urb_2 | $P_2$: 0.0134 (0.34) | $T_2$: 0.012 (0.30) |
| Urb_3 | $P_3$: 0.306 ** (2.48) | — |
| Constant | 0.453 *** (14.33) | 0.463 *** (14.42) |
| $R^2$ | 0.435 | 0.405 |
| Ajusted $R^2$ | 0.416 | 0.387 |
| F | 23.15 | 22.12 |
| N | 436 | 436 |

Standard deviations are provided in parentheses. ***, **, and * represent significance at the 1%, 5%, and 10% levels, with the 5% level being the commonly used significance level.

In $P_2$, which mainly covers counties in Shanxi, Shaanxi, Gansu, and Sichuan, the seasons are well-defined, with cold winters, hot summers, and often a rainy season in the Huang–Huai–Hai warm temperate zone, where temperatures exceed 30 °C in July and August. Here, the expansion of urbanization has led to a reversal in its impact on forest ecological function, resulting in a promotional effect with a coefficient of 0.0134 (not statistically significant).

In $P_3$, mainly comprising counties along the southern YRB, the expansion of urbanization enhances the promotion effect on forest ecological function, with a significant coefficient of 0.306. These areas experience a significant Northern Hemisphere subtropical monsoon climate, with high-pressure systems forming in the interior of northwest Asia during the winter, leading to prolonged cold north winds. The favorable climatic conditions endow these areas with the capacity for forest growth and self-repair. In these regions,

the progress of urbanization contributes to the improvement of forest ecological function, indicating a higher degree of coordinated development between the two.

Similarly, when taking "Temp" as the threshold variable (Table 6, Column 2), the coefficients of "Urb" on "Eco" shifted from negative to positive as temperature increased. However, the absolute value of these coefficients did not exhibit a clear trend. With a single threshold value of 10.105 °C, within the resulting subintervals, $T_1$ and $T_2$, the coefficients were significant at −0.141 and not significant at 0.012, respectively. This suggests that in counties with higher annual average temperatures, factors other than urbanization may play a more significant role in influencing forest ecological function. The direction of the control variable coefficients aligns with the results in Table 6 (Column 1).

The coordinated development of urbanization and forest ecological function depended not only on the climate conditions of a county, but also on various natural factors, such as slope, elevation, and contiguous forest area, as well as human factors like economic development, population density, household income, and ecological engineering. Grassland restoration has squeezed the space for forest growth and inhibited the enhancement of forest ecological function. Conversely, increased afforestation and wetland restoration efforts have had a positive impact on forest ecological function.

## 4. Discussion

### 4.1. The Role of Climate in Shaping Urb-Eco Relationship

Coupling coordination is the result of the comprehensive scores of subsystems and the comprehensive effects of coupling degree [57]. Despite the urbanization development in the YRB promoting economic growth and improving people's living standards, it has also caused substantial harm to the forest ecosystem, leading to the relatively poor overall coordination between the two. The positive correlation between Urb and Eco coordination and both categories of climate factors suggests that a warm and humid climate is conducive to the harmonious development of urbanization and forest ecological function.

On a deeper dive, the explanatory role of climate in the relationship between urbanization and forest ecological function was examined. There are two structural change points for the threshold variable "Pre" in the context of Urb, located at 532.342 mm and 694.178 mm, respectively, which are reasonable, because, according to China's precipitation classification, these thresholds fall within the semi-humid range, leading to the division into 3 distinct zones: $P_1$, $P_2$, and $P_3$. Additionally, when considering "Temp" as the threshold variable for Urb, there exists only a single threshold value at 10.105 °C, resulting in categorization into two zones: $T_1$ and $T_2$. To be specific, firstly, the direction of Urb's impact changes from negative to positive, suggesting that the detrimental effect of urbanization on forest ecological function tends to manifest in colder and drier counties, while its beneficial effect tends to manifest in warmer and wetter counties. Secondly, for counties with lower ($P_1$) and higher ($P_3$) humidity levels in the basin, urbanization has a greater impact on the forest ecological function. This pattern is not easily discernible when taking annual average temperature as the threshold variable since there is only a single threshold. Thirdly, a comparison of the threshold effects of the two climate factors, Temp and Pre, shows that Urb's impact is much more sensitive on Pre. This suggests that urbanization development is driving the transformation of agricultural land into urban land, the migration of surplus rural labor to the urban areas, industrial optimization and upgrading, as well as the expansion of green spaces. The release of ecological space signals a positive contribution to the enhancement of forest ecological function. The results indicate that climate explains the spatial differences in the impact of urbanization on forest ecological function in the YRB.

### 4.2. Mechanism

This study explored the intricate interplays between climate and nonlinear impacts of urbanization on the forest ecological function. The results show that climate plays a moderating role in the effects of urbanization on the forest ecological function, manifesting a trajectory that transitioned from constraint to facilitation. Specifically, in the process of

urbanization affecting forest ecological function, a dual threshold effect emerged concerning annual average precipitation, while a single threshold effect was observed for annual average temperature. A climate threshold framework was subsequently established based on these results.

Favorable water and thermal conditions serve to mitigate the adverse consequences of urbanization on the forest ecological function. In regions with a more humid and warm climate, urbanization's influence on these functions shifts from constraining to facilitating, in stark contrast to the arid and cold climate regions. This transformation can be attributed to two positive influences on the forest ecological function: the regulatory impact of superior thermal and hydrological conditions, and the ameliorating effect of heightened urbanization. Specifically, warmer and wetter climate effects promote forest growth and ecosystem resilience, as substantiated by this study. It underscores the role of hydrothermal conditions in shaping the extent to which urbanization impacts forest ecological function. Areas with favorable precipitation conditions generally exhibit superior vegetation growth dynamics and restoration capacity compared to areas with insufficient precipitation [58]. Over shorter timeframes, seasonally arid forests exhibit lower resilience than their humid tropical counterparts [59]. Yao et al. (2021) explored the agro-pastoral zones experiencing significant land use changes, whereby ecosystem resilience was significantly enhanced in wetter environments but weakened in drier ones [60]. Moreover, climate influences the forest ecological function by affecting forest species distribution [47], forest structure [49], forest productivity [48], plant functional traits [50], and forest ecological services [51,52]. As forests progressively assume the role of ecological "reservoirs" under climate regulation, their capacity to self-repair and accommodate human-generated emissions and pollutants increases [61]. This aids in mitigating the adverse impacts of urbanization on forests.

On the flip side, urbanization contributes to the enhancement of forest ecological function. Following the classic Environmental Kuznets Curve theory, there exists an "inverted U-shaped" relationship between economic development and environmental quality. Once a certain threshold is surpassed, environmental quality improves as economic development levels rise due to the predominance of positive effects over negative ones [62]. This is exemplified by the "promotion effect" of economic development on the ecological environment [38]. Wen et al.'s (2018) research aligns with the outcomes of this study, demonstrating that precipitation plays a significant positive moderating role in the urbanization process affecting vegetation cover [16]. Urbanization accompanies the influx of numerous production factors, leading to increased production efficiency and the transformation of various forest resource types [4], thereby augmenting centralized environmental management [7]. Furthermore, as urbanization progresses, both the government and public awareness of environmental protection intensify, benefiting the enhancement of forest ecological environments and functions. Li et al. (2021) reported that the implementation of reforestation projects could diminish forest fragmentation [17]. Ehrhardt-Martines et al. (2002) also noted a decreasing rate of natural resource consumption, including forest land, due to urbanization [12].

## 5. Strengths and Limitations

### 5.1. Strength of the Study

This study offers a proactive roadmap for addressing the challenges posed by global climate change and achieving a harmonious balance between urban development and environmental conservation. It serves as an innovative reference for the development of sustainable and climate-resilient urbanization policies. The main differences between our study and previous studies are as follows.

Firstly, our study introduces a climate perspective to investigate how climate affects the relationship between urbanization and forest ecological function. Previous studies often used urbanization itself as a threshold variable to explore the nonlinear impact of urbanization on the ecological environment under the different urbanization gradients [63–66]. In our study, climate factors serve as threshold variables, with urbanization as the threshold-

dependent variable, allowing for an examination of the potential moderating effect of climate on the nonlinear impact of urbanization on the forest ecological function.

Secondly, previous studies predominantly relied on the urban population proportion (Urb) as a representation of urbanization indicators, often overlooking a comprehensive assessment of urbanization levels [43]. In contrast, our study not only considered the transition from agricultural to urban populations, but also incorporated various aspects of urban development, including changes in land use, adjustments in industrial structure, and modifications in urban green coverage. Consequently, it introduced variables, such as land urbanization, social urbanization, spatial urbanization, and ecological urbanization to construct a comprehensive urbanization index. On the other hand, compared to using forest coverage as a single proxy indicator of forest quality, the forest ecological index can more accurately and comprehensively reflect the structure, ecological functions, and overall efficiency of the forest [67,68]. Therefore, our study adopts a more comprehensive forest ecological functional indicator.

Thirdly, our study focuses on the county-level scale, as this is a crucial governance unit that bridges urbanization and ecological civilization construction. However, research at the county level and even smaller scales has been limited due to a lack of ground-based experiments [14]. Through county-level research, we have established a climate threshold system that can characterize how urbanization affects forest ecological function, and our results will be instrumental in offering decision support to the management authorities.

*5.2. Limitation of the Study*

Firstly, this study employs cross-sectional data and does not delve into the dynamic changes of variables. Forest ecological function is influenced by both natural and human factors. Natural factors exhibit long-term stability, whereas human activities, such as socio-economic factors, are more dynamic and constitute essential components of current research on the vegetation cover impacts [69]. Thus, this study considers it feasible to investigate climate distribution differences within the same year. Secondly, this study compensates for missing data by incorporating data from other counties within the same city, which might have introduced the disparities between the estimated values and actual values. In future research, the fuzzy numbers can be employed to represent the uncertain data [70], facilitating further exploration of the impacts of different types of urbanization on the forest ecological function. Thirdly, we have investigated only two primary climatic factors, temperature, and precipitation. Thus, future research needs to include other climatic factors. Fourthly, in exploring the impact of urbanization on the forest ecological function under the climate context, we chose to utilize data at an annual scale. In future studies, the consideration of data at a finer scale, such as quarterly or monthly intervals, can allow for a more precise and comprehensive examination of the effects of climate-shaped urbanization on the forest ecological function. Lastly, the threshold values of temperature and precipitation were not considered as fixed as they are subject to variations depending on the study year, study area, and selection of indicators. Within each climate gradient, the impact coefficient of urbanization on forest ecological function will also change.

## 6. Conclusions and Recommendations

This study, based on the county-level data acquired from the Yellow River Basin in China, examined the shaping role of climate factors in the relationship between urbanization and forest ecological function. With this motivation, the research unfolds along two lines. The first line of inquiry employs a coupled coordination modeling to qualitatively assess the coupling and coordination between urbanization and forest ecological function, as well as their connection to climate factors, temperature, and precipitation. The second line of inquiry utilizes a threshold regression model that estimates how the impact coefficient of urbanization on forest ecological function varies under the different precipitation and temperature gradients. This approach aims at constructing a climate threshold system that elucidates the impact of urbanization development on the forest ecological function.

The major conclusions are: (1) Both the systematic coupling degree and the coupling coordination degree between urbanization and forest ecological function exhibit positive correlations with two climatic factors: annual precipitation and annual temperature. This suggests that a warm and humid climate facilitates the coordinated development of urbanization and forest ecological function. (2) The spatial differentiation in the impact of urbanization on the forest ecological function can be explained by climate. Urbanization tends to exert a stress effect on the forest ecological function in colder and drier counties. Conversely, in warmer and more humid counties, the dominant role played by the "reservoir" effect of forest ecological regulation, which mitigates the stress effects of urbanization. Consequently, the promotional effect of urbanization becomes apparent. In counties with the lowest and highest humidity levels, urbanization has a greater impact on the forest ecological function. Since there is only a single threshold, this pattern is difficult to capture when annual temperature is used as the threshold variable. The impact coefficient of urbanization on the forest ecology is more sensitive to precipitation than temperature.

This research provides a fresh perspective and theoretical framework for tailoring region-specific strategies in managing the pace of urbanization, bolstering early warning mechanisms for forest ecological function, and crafting conservation strategies. These findings yield two notable policy recommendations. Firstly, by categorizing urban planning based on the precipitation and temperature gradients, it becomes feasible to effectively oversee the urbanization development, ensuring the harmonious coexistence of urbanization and forest ecological function. In regions marked by conspicuous stress effects induced by urbanization, it becomes imperative to rigorously control the pace of development, enforce stringent regulations concerning land use transitions, mitigate the rapid expansion of urban populations, carefully reconfigure the industrial structure, and direct capital and technological investments towards urban core areas to enhance land use efficiency [71]. This approach effectively mitigates resource misallocation. In areas where the beneficial impacts of urbanization become discernible, urbanization can be incrementally advanced, but it should refrain from hastily expanding towns and launching construction initiatives [16]. Moreover, it is vital to fortify ecological safety measures and nurture ecosystem development [5]. Secondly, it is crucial to continue promoting urban forest development while maintaining a balance between socio-economic growth and forest ecosystem construction. This entails the establishment of an integrated urban–rural forest ecosystem [72]. Implementing afforestation and wetland restoration projects plays a pivotal role in enhancing forest ecological function. Consideration should be given to creating peri-urban ecological buffer zones, primarily composed of forests and wetlands, in suburban areas. These zones should serve as the central framework for a network of forests and water bodies, fully realizing their ecological potential.

**Supplementary Materials:** The following supporting information can be downloaded at: https://www.mdpi.com/article/10.3390/land12112047/s1, Table S1: cover area by each climate zone and its proportion in the Yellow River Basin; Table S2: forest ecological function subsystem indicators. References [37,67] are cited in the supplementary materials.

**Author Contributions:** Conceptualization, X.Z.; funding acquisition, X.Z.; writing—original draft, X.G.; writing—review and editing, G.W., R.P.S. and S.Z.; formal analysis, L.F. (Liyong Fu), X.G. and X.Z.; data curation, L.F. (Linyan Feng), Q.W., S.Z., Y.D. and X.G.; visualization, H.Z. All authors have read and agreed to the published version of the manuscript.

**Funding:** This research was funded by National Natural Science Foundation of China: "Evaluation of Site Quality and Productivity Estimation of Natural Forests Based on Forest Biomass" (No. 31971653); the China Scholarship Council: "Chinese Government Scholarship" (No. 202203270009); the Fundamental Research Funds of IFRIT: "Key Technologies for Accurate Extraction of Tree Growth Phenotypes" (No. CAFYBB2022ZB002); and Land Greening and Ecological Restoration Management Projects of National Forestry and Grassland Administration under Grant: "Vegetation Suitability Evaluation in the Yellow River Basin" (No. 213205).

**Data Availability Statement:** The data presented in this study are openly available at https://data.cnki.net; https://www.resdc.cn/Data.Search.aspx, accessed on 24 September 2023.

**Acknowledgments:** We are indebted to the anonymous reviewers and editor.

**Conflicts of Interest:** The authors declare no conflict of interest.

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
