# Peer review of "Relationship between Climate-Shaped Urbanization and Forest Ecological Function: A Case Study of the Yellow River Basin, China"

_land, doi:10.3390/land12112047_

Round 1

Reviewer 1 Report

Comments and Suggestions for Authors

A few comments and remarks on the article Gu et al. “Relationship between climate shaped-urbanization and forest ecological function: a case study of the Yellow River Basin, China”.

“…forests exhibit greater sensitivity to urbanization in regions with extreme climates.” (Lines 29-30).

What exactly do the authors mean by "extreme climate"? What criteria were used to identify climate extremes? This is absent from the manuscript. Please explain.

Forests in urbanised regions (in cities) constitute a unique type of ecosystem. For urban forests (green stands), in addition to the typical characteristics used for natural forests, it is normal to evaluate additional parameters. For example, their life status, because of road traffic and enterprises impacts forests. With “equal parameters” between natural and urban forests, their life state will affect their ecological functions. In addition, the accessibility of urban forests to urban residents impacts their ecological functions. Whether these parameters have been considered in the assessment of forest ecological functions?

When analysing forest ecological functions, was the existence or lack of nature reserves and national parks in the research region considered? Their overall number, size, and proportion to the entire study area? The extent of forest ecological functions is also greatly influenced by these variables.

The subsection Statistical analysis is missing from the Materials and Methods section. Please detail the procedures used to handle the data in a statistical and mathematical manner.

“Using ArcGIS software…” (Line 260), “Stata 17.0 was employed to…? (Line 326), etc.

In the Materials and Methods section, you must list all the software tools used, including their names, e-references, procedures, and algorithms.

“The climate threshold values and impact coefficients in the urbanization effect on forest ecological function may change accordingly.” (Lines 575-576).

How much of a change in the evaluation of forest ecological functions in the study region can be attributed to variations in the climatic characteristics?

Reviewer 2 Report

Comments and Suggestions for Authors

This is an interesting paper. It successfully integrated ecological and sociological data of YRB and explored the effects of climate on urbanization and forest ecological function, successfully. Data analysis seems relevant overall and outputs are significant. 

However, I have little agreement on whether the study design is relevant to prove the Environmental Kuznets Curve theory. The Environmental Kuznets Curve can be indicated by the pattern between economic development and environmental properties when other factors are controlled including environmental and climate factors. However, the gradient of ecological function, which is assumed as a dependent factor in the theory, is not only affected by not only economy, but also climate. This is exactly the “relationship between climate shaped-urbanization and forest ecological function”, which is entitled by the authors; however, it may differ to the Environmental Kuznets Curve theory. If my understanding is something wrong, please strengthen the issue in the manuscript. In my opinion, the manuscript is great enough even if it doesn’t adopt the Environmental Kuznets Curve theory. 

Minor comments. 

L.65 The acronym “EKC” should be defined when it is introduced in the main text first, whether it is defined in abstract. Overall, please carefully double check the use of all acronyms. Some of their definitions are missed or presented at a wrong place. 

L.72 Could you introduce an example from a temperate region?

L111 Could you briefly add some explanation for the definition or the source of the climate zone classification?

L113 A flowchart of data analysis will be helpful to readers. The analysis units and data are shaped in polygon, not raster. I agree with the reasoning, but some readers may question the non-raster-based spatial analysis. Can you justify this?

Table 2 The descriptive statistics can be presented for each of 8 climate zones, which are classified in the Fig. 1b. this can be presented in the main text, the appendix, or the supplementary. In addition, the cover area of each climate zones can be included. 

L147 It must be Table 1, not Table 2. Please check the numbering of figures and tables. The acronyms of C and D should be defined in the table independently from the main body. Check the use of acronyms in the tables and figures.

L.191 How were the Forest Ecological Function Index (Eco) and the urbanization variable (Urb) evaluated? Has the methodology been validated? More details should be given in the main text or the supplementary.

L 299 The terminology of the Eco-Urb is inconsistent among Figure 3, materials and methods, and results section. 

L.354 What does the “g1” indicate in Table 4?

L 366 What do the “Pre, Pre 1st round, and Pre 2nd round” indicate? In addition, is there a particular reason not to harmonize the scale of y axis?

L 415 As a non-Chinese reader, I feel difficulty in following the discussion section, because most of comparison was made with Chinese cases? The China-oriented study must be meaningful, nevertheless, please don’t forget potential global readers. 

Reviewer 3 Report

Comments and Suggestions for Authors

1.The article studies the relationship between climate based urbanization and forest ecological function, which is interdependent. However, the literature review only summarizes the impact of urbanization on forest ecological function. It is recommended to organize and supplement the research status that is more in line with the research content.

2. Line36-41: In the introduction section, it is recommended to provide more detailed causal relationships. Please supplement and improve the specific reasons for conflict between forest resource conservation and urban expansion

3. Lines60-62, please explain why the positive promoting effect is not the same as the negative promoting effect

4. Lines70-77: Previously, there was no explanation of human relationships, and causal relationships do not hold true

5. Line99: The main land use types in the Yellow River Basin are grassland and arable land, while forests are not the main land use types. Why was it chosen as the research object? Suggest adding a spatial distribution map that reflects the forest and urban construction land in the article.

6. Line109: Figure 1. c suggests adjusting by placing "Annual average precision" above the specific precipitation legend for readers to understand.

7. Line109: The image is not aligned, and the grid of longitude and latitude in Figure b and Figure d overlaps. Consider aligning the four images with the grid of longitude and latitude, and aligning the north arrow separately, as shown in Figure 2, etc

8. Line101: Yellow River Basin specific atmospheric circulation and monsoon conditions to illustrate the impact of these factors on the research in the following text

9. Line111: The drawing number is bold, but the drawing name is not bold. Please unify it as 264.300.311.323. 368

10. Line121: Line127, Line135, first, second, third unified

11. Line121-146: It is recommended to simplify the content of coupling coordination model calculation, and add appropriate textual explanations based on listing formulas and clearly explaining the meaning of parameters. The current description is a bit lengthy.

12. Line204: "Threshold dependent variable" does not start with "2.3.1 It is recommended to add bullet marks to other variables in the "Indicator selection" section.

13. Line214-217: Is there any previous research or literature providing support for the alternative variables you have selected to represent URLs, URPs, URIs, and URS?

14. Line218-225: It is recommended to supplement the role and importance of hydrothermal conditions on urban systems and forest ecosystems.

15. Line226: "We will also control..." It is recommended to modify the tense to the present perfect tense. Please pay attention to the tense of the sentence.

16. Line243: Is it scientific to represent the climate status of a region with annual average precipitation and temperature? Do you think the two regions have similar climates due to different months of concentrated precipitation, but similar annual precipitation?

17. Line257: Can the note section above "In the wetland restoration domain, major projects include the grain for green project and wetland protection and restoration project" explain in detail why the wetland restoration field mainly includes returning farmland to forest projects?

18. Line262: The top of figures 2 (a) and (b) are not aligned, it is recommended to adjust.

19. Line264: Figure 2 Legend: Why are the two legend sequences reversed

20. Line324: Suggest supplementing references related to threshold testing

21. Line394: This paragraph discusses how favorable climate promotes forest growth and restoration, but how does urbanization help improve forest ecological functions? Please provide additional analysis on this conclusion.

22. Line468: "On the other hand, urban expansion has led to the widespread conversion of forest land into different land use types, resulting in various forest zones within forest resources. This in turn provides a broader ecological space for forests." It is recommended to provide a specific explanation for this.

23. Line495-525: It is recommended to add specific research results or literature from other scholars to compare with other studies, and provide a detailed explanation of the innovative aspects of your research.

24. Why are the serial numbers after Line527:4.4 4.3.1 and 4.3.2

25. Line542-561: It is recommended to add relevant suggestions and development strategies for urban development and forest ecology in this research area.

26. References: The format of individual references is inconsistent. Please note whether the format of the references meets the requirements of the journal.

Comments on the Quality of English Language

Moderate editing of English language required

Round 2

Reviewer 1 Report

Comments and Suggestions for Authors

I must acknowledge that the authors have made an impressive effort in taking into account all the suggestions I made in my revission. In its current state I consider that the manuscript has been reached the publishing level and can be accepted.

Reviewer 2 Report

Comments and Suggestions for Authors

All responses and revisions made by the authors are clear, thoughtful, and satisfactory. I am gaining a lot of learning and insights from the authors' works.  Thank you very much.

Reviewer 3 Report

Comments and Suggestions for Authors

Accept in present form

Comments on the Quality of English Language

Minor editing of English language required